# Secure Networking with Software-Defined Reconfigurable Intelligent Surfaces

**DOI:** 10.3390/s23052726

**Published:** 2023-03-02

**Authors:** Francesco Chiti, Ashley Degl’Innocenti, Laura Pierucci

**Affiliations:** Department of Information Engineering, University of Florence, 50139 Firenze, Italy

**Keywords:** reconfigurable intelligent surfaces, software-defined networking, physical layer security, secrecy capacity optimization

## Abstract

Reconfigurable intelligent surfaces (RIS) are considered of paramount importance to improve air–ground and THz communications performance for 6G systems. Recently, RISs were proposed in Physical Layer Security (PLS), as they can (i) improve the secrecy capacity due to the controlled directional reflections’ capability of RIS elements and (ii) avoid potential eavesdroppers, redirecting data streams towards the intended users. This paper proposes the integration of a multi-RISs system within a Software Defined Networking (SDN) architecture to provide a specific control layer for secure data flows forwarding. The optimisation problem is properly characterised in terms of an objective function and an equivalent graph theory model is considered to address the optimal solution. Moreover, different heuristics are proposed, trading off complexity and PLS performance, to evaluate the more suitable multi-beam routing strategy. Numerical results are also provided, focusing on a worst case scenario which points out the improvement of the secrecy rate from the increase in the number of eavesdroppers. Furthermore, the security performance is investigated for a specific user mobility pattern in a pedestrian scenario.

## 1. Introduction

Recently, Intelligent Reflective Surfaces (IRS) have been consideredof paramount importance for beyond 5G and 6G cellular systems, [1] in order to increase the link efficiency in terms of data rate, coverage, and connectivity, as well as being a good means to reduce energy consumption. Since one of the most relevant challenges for 6G are the air-to-ground communications, IRSs can also be employed to enlarge the cellular coverage provided by aerial platforms or the unmanned aerial vehicle, and consequently to optimize the air-to-ground average rate [2], as well as in conjunction with the non-orthogonal multiple access (NOMA) technique for the uplink cellular network [3]. Another key enabling the technology for 6G is Terahertz (THz) communications. IRSs applied in THz systems can dynamically control the propagation of THz signals and overcome the signal communication disruption due to obstacles by steering the THz signals in the desired direction, or, they can act as relays to increase the coverage area [4]. Specifically, IRS is a planar array with a high number of passive small reflecting surface/elements, called tiles, which allow one to change the amplitude and phase of the incident electromagnetic wave in order to reflect it passively in the desired spatial direction, mainly towards the receiver or another IRS, in a reconfigurable way [1,5].

This vision paves the way towards the related concept of the reconfigurable intelligent surface (RIS) [6], where a software controller can jointly optimise and configure the input and reflecting waves based on sensed data. Furthermore, as RISs can interact, reconfigure and program the wireless environment, they can be adopted to enhance the Physical Layer Security (PLS). Specifically, RISs can improve the secrecy capacity of the channel, by allowing data streams to reach the intended users, avoiding eavesdroppers as much as possible by means of directional reflections, especially for the Internet of Things (IoT) scenario [7].

Considering an IoT domain, the novel concept of Software-Defined Networking (SDN) allows the reactive control of traffic flows by configuring the network resources to meet the application requirements [8]. Indeed, SDN is a networking approach that decouples the *control* of the network resources from the operations related to the information *forwarding*. Through specific software-based SDN Controllers, the entire network could be centrally managed with an improved efficiency thanks to the separation between the control plane and hardware devices (data plane). Moreover, SDN combines the benefits of the virtualisation and compute continuum, thus allowing the definition of new intelligent architectures that support more complex network functionalities to offer a more advanced service, e.g., through the integration of wired and wireless systems. According to this promising vision, the SDN architecture has been recently proposed to enhance the PLS, where a SDN Controller is able to perform a device selection in order to optimise the secrecy capacity [9].

To address the above issues, this paper introduces a SDN Controller in a RIS-aided system to dynamically optimise the path between the source and destination (legitimate user) through multiple RISs in order to avoid malicious nodes, where we assume RISs as secure nodes. Differently from other papers in the literature, which analyze attacks to the RIS controller, we consider uplink communications from legitimate users to the base station (or access point) where the eavesdroppers want to replace legitimate transmissions, therefore considering the worst case scenario due to the lower uplink transmission powers. According to the multiple RIS approach, each authorised network device cooperates for a common goal (i.e., multi-hop cooperative RISs), with the advantage of increasing the power gain of $O(M^{4})$, where *M* represents the number of reflecting elements, which exceed the value of $O(M^{2})$ related to the single RIS reflection link, as discussed in [10].

Specifically, graph theory methodologies have been applied for designing a PLS anti-eavesdropping strategy. In particular, a weighted graph is derived from the network topology, where a Shortest Path Algorithm (SPA) is applied to evaluate the appropriate path in order to connect the source and destination node, with the aim of improving security metrics.

Furthermore, to the best of our knowledge, the use of multiple RIS-assisted system applied to the case of mobile users has not been investigated before. This paper presents, indeed, a mobility pattern in a pedestrian scenario, with a limited area (i.e., such as a square) being representative of a realistic case study occurring in an urban scenario to test the security related performance of the proposed integrated multiple-RISs and SDN approach.

The most relevant challenges of this paper can be detailed as follows:We design a cooperative multi-beam multi-hop routing strategy that involves multiple IRSs to select the best reflection path between source and destination to improve PLS performance also in the case of mobile IoT devices with moderate speed;We analytically derive the overall end-to-end (e2e) sum-secrecy rate maximisation problem subject to the constraint for assuring a target capacity at the destination;We propose different heuristics to avoid eavesdroppers with a reasonable complexity;We propose the integration of IRSs into an SDN architecture to address the implementation of the proposed framework.

The paper is organised as follows. Section 2 reviews some papers related to RIS technologies and their application towards PLS. Section 3 presents a comprehensive model for the considered secure multiple RIS-aided IoT system, along with the related SDN architecture. In addition, the optimisation problem is presented, together with the low complexity heuristics. Then, Section 4 shows the numerical results in terms of the average secrecy rate for different eavesdroppers’ deployments. Section 5 addresses our main results and indicates the future directions for a secure multi-RIS system.

## 2. Related Work

Several comprehensive surveys on the emerging RIS technology are provided in the literature, e.g., in [5,11], focusing on potential applications, challenges, hardware architecture and practical constraints, while highlighting its ability to control and manage the wireless channel to enhance the communication performance.

The authors in [12] enlarge the perspective of a programmable wireless environment by providing a review on the impact of double or multiple RIS use. First, they analyse the advantages of multiple RISs with respect to single-reflection links and, then, highlight two main issues: (i) the channel state acquisition with a low overhead in the case of bad channel conditions due to a large number of obstacles and (ii) the multi-RIS selection optimisation for each user to maximise the total throughput and narrow interference among users.

In the existing literature, several papers usually consider attack models directly to the controller of RIS systems, which provides decisions on the spatial reflections of the signals, therefore increasing the vulnerability of the received signal, as in [13,14,15]. The vulnerability of pilot signals is considered, where pilot signals are transmitted to alter the channel state estimation to the legitimate users, i.e., the pilot spoofing attack. In [16], the RIS is used to improve the security of traditional key generation technology. An RIS-assisted Manipulating attack (RISM) is considered, where a malicious user can destroy the uplink and down link channel reciprocity. Then, a slewing rate detection method was proposed which can detect and separate the attacked path, maximising the key generation rate.

A classical approach to improve PLS is investigated in [7], where a scenario with an access point (AP) aided by RIS, a legitimate user and one eavesdropper, was evaluated. The paper provides the well-known result that the secrecy rate was improved by increasing the number of RIS elements and decreasing the average signal-to-noise ratio (SNR) that was closer to the eavesdropper.

In another contribution, RISs are used for enhancing the PLS of wireless communications systems [17] and, in particular, the classical scenario, with an RIS assisting a multi-antenna transmitter to secure the wireless system, with a single-antenna receiver and one eavesdropper, is adopted. The block coordinate descent (BCD) and minorisation maximisation (MM) techniques are developed to solve the non-convex optimisation problem related to the joint optimisation of the beamformer at the transmitter, and the RIS phase shifts to maximise the secrecy rate.

The authors in [18] consider the problem of the secrecy rate maximisation in the case of eavesdroppers equipped with a multiple-input multiple-output multi-antenna (MIMOME) and an RIS-assisted Gaussian wiretap channel (WTC). In particular, the block successive maximisation (BSM) method is adopted which considers a lower bound on the secrecy rate to optimise the input covariance matrix, while the maximisation is carried out for each individual phase shift.

In [19], the secure communication is analyzed in a STAR-RIS configuration, i.e., where the source and legitimate user cannot be at the same side of the RIS, and integrated with the non-orthogonal multiple access (NOMA) strategy. Furthermore, an artificial noise (AN) was introduced to improve the secure communications. The non-convex optimization problem related to AN and NOMA parameters and the number of RIS elements to improve the secrecy rate is solved by using the successive convex approximation (SCA) and semi-definite relaxation (SDR) techniques.

In [20], the authors consider a multiple antenna system at the transmitter side, and include a legitimate user and an eavesdropper. To secrete the information, the RIS reflection coefficients are weighted with multiplicative random coefficients in each transmission, but the channel matrix is diagonally maintained for the intended user. As a consequence, the legitimate user can decrypt the information, while the eavesdropper cannot decode it. The paper suggests three different reflection randomness schemes to diagonalize the channel matrix by using: (1) the space-shift keying (SSK) modulation, (2) quadrature space-shift keying (QSSK) modulation and forcing the diagonal elements to have real values, (3) multiple phase-shift keying (PSK) and converting diagonal elements to assume real values again.

Differently, the paper [21] suggests that the induced randomness on the RIS reflection coefficients could also be performed on the signal received by the eavesdroppers and consequently they can collaborate to estimate the RIS-generated random channel and decode the secret key.

Finally, as discussed in [19], RISs are integrated in an SDN architecture to enforce the advanced PLS, with the main goal of detecting the position of possible eavesdroppers and steering the wireless waves in the space to avoid being close to malicious users through the orchestration of a set of metasurfaces.

In this paper, we consider an integrated SDN-RISs architecture as in [19], where multiple RISs dynamically act as relays to support e2e data flow delivery with the aim of avoiding eavesdroppers present in the path. However, differently from [19], which only shows the feasibility of an integrated reconfigurable antenna system with SDN, (i) we have proposed several heuristics with increasing computational complexity to optimize the multi-beam routing strategy focused towards PLS and (ii) we demonstrated the simulation results related to the secrecy rate performance also considered users’ mobility.

In conclusion, to the best of our knowledge, the use of SDN and the multiple RIS-assisted system applied to the case of mobile users has not been investigated before in the literature.

## 3. System Model

In deriving a suitable model for investigating the proposed approach performance, we consider an outdoor scenario comprised of K+2 legitimate nodes, where a source $A_{0}$ transmits to a destination $A_{K+1}$ (e.g., the base station), on a path $\Omega =(A_{0},A_{1},...,A_{K+1})$ formed by *K* RISs and a number of $E_{N_{e}}$ eavesdroppers, as pointed out in Figure 1.

In particular, we propose the integration of multiple RISs into a SDN general framework. To this purpose, we refer to the classic architecture composed of the Application, Control and Data Planes (AP, CP, DP) already proposed in [18], but focused on security rather than mere communications. Specifically, RISs belong to DP and in turn are connected to the SDN Controller via a proper gateway, where a southbound interface is implemented to allow a bidirectional control message exchangein order to monitor and control the e2e data forwarding. As this connection is wired, the adoption of the standard OpenFlow protocol can be assumed [22].

Moreover, the following operations were considered in Figure 1 as part of the Controller management logic:*Network Discovery* and *Tracking*: responsible for the device location through an algorithm of the compressed sensing of wavefronts involving RISs [23];*Device Access Control*: intended for classifying the users;*Topology Manager*: responsible for maintaining an accurate abstract representation of the network;*Decision Engine* and *Path Selection*;*RIS Link Deployment* and *End-Device Beamforming*.

It is worth noticing that these functionalities are based on information collected by the SDN Controller via proper messaging and processed by a specific algorithm in order to derive the global network state. Figure 1 also presents the steps in orchestrating the overall control workflow performed by the SDN Control:Channel state information (CSI) sensing;Network discovery (usually represented as an annotated graph) and device tracking;Device access control;Topology management and optimum path selection;RIS link set-up and end-device beamforming;e2e data flow transmission;Device configuration.

The SDN Controller abstracts the network topology into an anti-eavesdropping weighted graph, where the legitimate nodes are part of the vertex set and the links between them are part of the edge set. Usually, RISs can provide additional paths as an alternative to the direct path, in the case where it is not available due to obstacles and/or poor channel quality. In general, the e2e capacity of a source to the destination flow over a given path $\Omega =(A_{0},A_{1},...,A_{K+1})$ is given by:(1)$$C_{\Omega }^{d}=\log_{2}\left(1+\gamma_{d}\right)\;\left[\mathrm{bps}/\mathrm{Hz}\right]$$
where $\gamma_{d}$ is the received signal to noise ratio (SNR) at the destination. Similarly, the e2e capacity of the wiretap channel can be detailed as follows:(2)$$C_{\Omega }^{E}=\log_{2}\left(1+\gamma^{E}\right)\;\left[\mathrm{bps}/\mathrm{Hz}\right]$$
where $\gamma^{E}$ is the e2e SNR for the eavesdropper.

If we consider the presence of eavesdroppers, whose goal it is to intercept the legitimate communications, it is necessary to select the path that maximises the secrecy rate, which is defined by [24] as:(3)$$C_{\Omega }^{S}=\max\left(C_{\Omega }^{d}-C_{\Omega }^{E},0\right)\;\left[\mathrm{bps}/\mathrm{Hz}\right]$$

It could be pointed out that a secure communication is possible if and only if the legitimate user experiences a greater channel capacity than the one of the eavesdroppers.

In a free-space propagation scenario, where an ideal isotropic transmit antenna transmits to a lossless receiving antenna located at distance *d*, then, it is well known that the received power is:(4)$$P_{rx}=\frac{A}{4\pi d^{2}}P_{tx}$$
where $P_{tx}$ is the transmitted power and $\beta_{d}=\frac{A}{4\pi d^{2}}$ is the free-space channel gain, or pathloss; $A=\frac{\lambda^{2}}{\left(4\pi \right)}$ is the area of an isotropic receive antenna.

First, we consider an uplink communication aided by one RIS. The RIS has *M* passive reflecting elements, also called tiles, that can reflect the incident signal in a reconfiguirable way, but without amplifying it. In particular, each tile controls the individual phase and magnitude of each reflected signal. We assume a line-of-sight (LoS) propagation.

The e2e-received signal at the destination can be modeled as:(5)$$r_{RIS}=g^{T}\Theta h\sqrt{P_{tx}}s+n$$
where *s* the source signal, and $n∼N_{C}\left(0,\sigma^{2}\right)$ is the additive white Gaussian noise (AWGN) at the receiver with variance $\sigma^{2}$. Moreover, $\Theta =diag(\mu_{1}e^{j\theta_{1}},...,\mu_{M}e^{j\theta_{M}})$ is the reflection coefficient matrix, where $\mu_{1},...,\mu_{M}\in \left[0,1\right]$ and $\theta_{1},...,\theta_{M}\in \left[0,2\pi \right)$ are the amplitude and the phase-shift variables, respectively. Moreover, $h={\left[h_{1},...,h_{M}\right]}^{T}$ with $h_{m}=\left|h_{m}\right|e^{-j\phi_{m}}$ represents the vector of the channel from the source to the RIS, while $g={\left[g_{1},...,g_{M}\right]}^{T}$ with $g_{m}=\left|g_{m}\right|e^{-j\psi_{m}}$ denotes the vector of the channel from the RIS to the destination. Finally, the channel capacity is given by:(6)$$\log_{2}\left(1+SNR_{RIS}\right)$$
where:(7)$$SNR_{RIS}=|g^{T}{\Theta h|}^{2}\frac{P_{tx}}{\sigma^{2}}=|\sum_{m=1}^{M}\mu_{m}|h_{m}\left|\right|g_{m}|e^{j(\theta_{m}-\phi_{m}-\psi_{m})}{|}^{2}\frac{P_{tx}}{\sigma^{2}}$$
as proposed in [25], where the *far-field approximation* approximation is also introduced. According to this, by assuming the source located at distance *d* in angle η and the destination located at distance δ in angle ω, if both the source and destination are lying within the far-field region of the RIS, or equivalently that $d\cos\left(\eta \right)>>\sqrt{MA}$ and $\delta \cos\left(\omega \right)>>\sqrt{MA}$, the SNR can be approximated as:(8)$$SNR_{RIS}=M^{2}\varsigma_{d,\eta }\varsigma_{\delta ,\omega }\frac{P_{tx}}{\sigma^{2}}$$
where:(9)$$\varsigma_{d,\eta }=\beta_{d\cos(\eta )}\cos^{3}\left(\eta \right)=\frac{A\cos(\eta )}{4\pi d^{2}}$$
and:(10)$$\varsigma_{\delta ,\omega }=\beta_{\delta \cos(\omega )}\cos^{3}\left(\omega \right)=\frac{A\cos(\omega )}{4\pi \delta^{2}}$$

### 3.1. Multi-RIS Setup

As previously introduced, we assume that multiple RISs are deployed to assist in the communication by providing different e2e LoS paths to connect the source and destination. However, this brings us to a trade-off: on one hand, each RIS in the path introduces a $M^{2}$ passive beamforming gain, on the other, it makes the multi-reflection pathloss term more significant. The received power over the considered path $\Omega =(A_{0},A_{1},...,A_{K+1})$ can be expressed according to [10], as:(11)$$|h_{A_{0},A_{K+1}}{\left(\Omega \right)|}^{2}=\frac{M^{2K}N\beta^{K+1}}{d_{A_{0},A_{2}}^{\alpha }d_{A_{K},A_{K+1}}^{\alpha }\prod_{k=1}^{K}d_{a_{k},a_{k+1}}^{\alpha }}$$
where:(12)$$\kappa \left(\Omega \right)=\frac{{\sqrt[{\alpha }]{\beta }}^{K+1}}{d_{A_{0},A_{2}}d_{A_{K},A_{K+1}}\prod_{k=1}^{K-1}d_{a_{k},a_{k+1}}}$$
is the pathloss term for a multi-RIS e2e path, and *N* is the number of antennas available at the BS, $\beta =\frac{A\cos(\omega )}{4\pi }$ and $A={\left(\lambda /4\right)}^{2}$.

### 3.2. Problem Statement

In order to optimise the physical security policy within an SDN framework, we resort to Graph Theory by relying on an equivalent shortest simple-path problem (SSPP), as proposed in [10] for a different objective function. As we are interested in finding the best e2e path that maximises the secrecy rate, the general optimisation problem (OP) can be formulated as:$\mathrm{Objective}\;\max_{\Omega }\;C_{\Omega }^{S}$;$\mathrm{Subject}\;\mathrm{to}\;\gamma_{d}\geq \gamma_{th}$;$\mathrm{Given}\;\left(x_{1},y_{1}\right),\left(x_{K+1},y_{K+1}\right),\left(x_{RIS_{i}},y_{RIS_{i}}\right),\left(x_{e},y_{e}\right)\;,\;\forall e=1,...,E\;,\;\forall i=2,...,K-2$.
where $\Omega =(A_{1},A2,...,A_{K})$ is the path that maximises the e2e secrecy rate. In the OP, we introduced the constrain $\gamma_{th}$, which is a target threshold that guarantees a sufficient channel capacity at the destination such that paths below that threshold are not taken into account.

We assume the SDN Controller is aware of the positions of legitimate users in the network and is able to estimate the presence and the positions of the eavesdroppers. This is performed by collecting (un)intended uplink signals received by IRS with a specific southbound interface control sub-protocol, storing it and further processing them with a generic localisation algorithm, e.g., a compressed sensing approach that is usually adopted in distributed passive sensing systems.

We derive a directed weighted graph $G^{E}=\left(V,E\right)$ that can abstract our network topology in the presence of eavesdroppers and apply a shortest path algorithm (SPA) to solve the OP. Then, the problem shifts to the formulation of an appropriate model for weights in the graph. In particular, the vertex set V is given by the legitimate nodes and the edge set E is represented by the links between them. The weight of the edge connecting the *i*-th and *j*-th vertexes is given by:(13)$$W_{i,j}^{E}=\xi *W_{i,j}+\hat{e}=\xi *\ln\frac{d_{i,j}}{M\sqrt{\beta }}+\hat{e}$$

Due to the complexity of the problem, which implies a high control overhead, we present five heuristics which simplify the weights $W_{i,j}^{E}$ to be applied in the derived graph; specifically, the heuristics are sorted by less to more computational complexity. In (Equation 13), the key idea is to balance the contribution of the distance/pathloss term (first term) and the impact of the eavesdroppers on each link ($\hat{e}$), while we propose the following heuristics:HEU1ξ=cost, $\hat{e}=cost=1\;\forall e\in E$ detected in the coverage area.HEU2ξ=cost, $\hat{e}=\rho \;\cdot \;\hat{X}\;\forall e\in E$ detected in the coverage area. ρ=cost and where $\hat{X}=\max_{i,j=1,...,|V|}W_{i,j}$, thus relating $W_{i,j}^{E}$ to the higher weight in $W_{i,j}\;\forall i,j=1,...,|V|$.HEU3ξ=cost, $\hat{e}=\frac{2\;\cdot \;\hat{X}}{\log_{10}\left(\max_{e}P_{i,e}\right)}$, where $\max_{e}P_{i,e},e\in E$ is the maximum value of the pathloss term in the channel between the transmitting node and each eavesdropper.HEU4ξ=cost, $\hat{e}=\frac{2\;\cdot \;\hat{X}}{|\log_{10}\left(\sum_{e=1}^{E}P_{i,e}\right)|}$, where $\sum_{e=1}^{E}P_{i,e}$ is the sum of the pathloss terms between the transmitting node and each eavesdropper.HEU5ξ=cost, $\hat{e}=\log_{10}\left(\sqrt{\sum_{e=1}^{E}P_{i,e}}\right)$.

Finally, we address two different operative scenarios:*Cooperative*: eavesdroppers are able to communicate with each other in order to combine the wiretapped signals and detect the legitimate user signals,*Non Cooperative*: eavesdroppers are independent and do not share information with each other.

As a consequence, the total wiretapped SNR over the path $\Omega =(A_{0},A_{1},...,A_{K+1})$ is, respectively:(14)$$\left(c\right)\;\gamma_{\Omega }^{E}=\sum_{r_{i,j}\in \Omega }\sum_{e=1}^{E}\gamma_{i,j,e}^{E}$$
(15)$$\left(nc\right)\;\gamma_{\Omega }^{E}=\max_{1\leq e\leq E}\sum_{r_{i,j}\in \Omega }\gamma_{i,j,e}^{E}$$

It is worth noting that in the cooperative scenario, eavesdroppers cooperate via exchanging messages, thus making it possible in principle for a SDN Controller to monitor, detect and localise them.

## 4. Numerical Results

In this section, we evaluate the performance achievable in the previously introduced scenarios to increase the physical layer security. In order to evaluate the secrecy rate performance, the numerical simulations are carried out by using the Matlab framework. We consider an outdoor multi-RIS system covering an area of size 100 m × 50 m with a single antenna transmitting source, one receiving Base Station with *N* = 32 antennas, eight RISs deployed in fixed locations and $E_{N_{e}}$ eavesdroppers randomly distributed.

As shown in Table 1, we assume $M=10^{3},P_{tx}=26\;\mathrm{dBm},\sigma^{2}=-94\;\mathrm{dBm}$, the carrier frequency is set at 6 GHz; the source is located at (40,10) m and the base station is located at (25,50) m. In all heuristics, we consider ξ=1, except in HEU4, where we set ξ=0.55 as a reasonable value in our approach. In HEU2, we consider ρ=0.25.Finally, we present both scenarios (c) and (nc). (a) represents the worst case scenario, since the eavesdroppers collaborate to combine the wiretapped signals and consequently $C_{\Omega }^{E}$ is larger, even though detection and location processes through the SDN Controller are more affordable.

In Figure 2, we consider the performance related to scenario (c). The average secrecy rate versus the number of eavesdroppers is shown according to the proposed heuristics and compared with (i) the optimal solution (exhaustive search) and (ii) the case where the basic SPA is applied to a graph only to maximise the e2e capacity (blind search). We do the same for scenario (nc) in Figure 3. It can be pointed out that the more complex the heuristics, the closer their performance is to the optimal solution, while HEU2 is similar to the blind search, which simply ignores the presence of eavesdroppers. As expected, the performances in scenario (c) are worse. In Figure 2, the gap between the optimal solution and the best proposed heuristics is larger then Figure 3. In this instance, eavesdroppers are not able to communicate with each other, so the wiretapped SNR will be smaller, benefitting the secrecy rate.

This result is confirmed by Figure 4 and Figure 5, which show the related secrecy rate loss with regards to the optimum performance, defined as:$$\frac{C_{opt}^{S}-C_{HEU_{i}}^{S}}{C_{opt}^{S}},i=1,2,...,5$$

Here, it is more evident that HEU4 is closer to the optimum one. In addition, as expected, in all of the considered cases, the secrecy performance decreases when the number of eavesdroppers increases. Moreover, it is worth noting that both HEU1 and HEU2 are computationally lighter than HEU3, HEU4 and HEU5, since they require fewer operations to evaluate the graph weights. Moreover, it is worth noting that both HEU1 and HEU2 are computationally lighter than HEU3, HEU4 and HEU5, since they require fewer operations to evaluate the graph weights. While HEU1 and HEU2 simply inspect the coverage area of a connection link for the presence of one or more eavesdroppers, the other heuristics build on this procedure. Therefore, they require more computational and storage resources. In particular, HEU1 and HEU2 do not require one to evaluate SNR for each eavesdropper, but only to estimate the presence of eavesdroppers in the link coverage area.

In order to correctly represent the network topology with a weighted graph, the SDN Controller needs to estimate the location of the eavesdroppers using a positioning protocol. To take into account the impact of this procedure, we introduce in the previous analysed case an uncertainty on the position of eavesdroppers modelled as a uniformly randomly distributed variable with a mean value equal to 1.5 m. In Figure 6, we examine the impact on the secrecy rate by considering five eavesdroppers: it can be noticed that reducing the accuracy on the localisation makes it harder to avoid the effect of the eavesdroppers, so that the overall performance worsens.

On a further note, to generalise the investigation of the security related performance, we introduced three concurrent mobile users in an urban scenario, whose mobility patterns are shown in Figure 7, where two of them move longitudinally (blue and green lines) and the other one follows a zig-zag trajectory (black line), all at a pedestrian speed. In addition to that, $E_{N_{e}}=20$ fixed eavesdroppers are introduced. To handle this case, we endow the SDN Controller with a scheduler module which adopts a weighted round robin policy; specifically, users are served by the SDN Controller in a static exclusive order (user 1, user 2 and user 3) and they cannot share any RIS within their e2e path. The average secrecy rate achieved by each user is, respectively, $C_{1}^{S}=10.8138,C_{2}^{S}=9.9317,C_{3}^{S}=9.9996$.

In Figure 8, the instantaneous secrecy rate is plotted as a function of simulation time; it can be pointed out that the scenario geometry and the position of eavesdroppers influence the performance. In particular, as one user passes close to an eavesdropper, the secrecy rate decreases, whereas its performance improves for more favourable positions along its path.

## 5. Conclusions

This paper focuses on integrating a multi-RIS system within an SDN-oriented architecture to improve the PLS performance via a secure data forwarding strategy. Specifically, the SDN Controller is able (i) to detect the presence of possible eavesdroppers and (ii) to perform the best path selection involving multiple RISs with the aim of improving the secrecy rate at the destination. We apply this framework to uplink communications from a source to a destination (mainly a base station), where multiple RISs are involved. We supposed that eavesdroppers adopt a collaborative approach to intercept the source signal; however, this allows the SDN Controller to detect their presence and, in turn, to react and dynamically select another path through the multiple RISs. To this goal, an equivalent graph theory model is first provided to evaluate the optimal solution, and, then different heuristics are proposed, trading off complexity and PLS performance, in order to evaluate the more suitable multi-beam routing strategy. Numerical results point out the effectiveness of the proposed approach by the increasing in the number of eavesdroppers. Furthermore, the security performance is investigated for a specific user mobility pattern in a pedestrian scenario.

## Figures and Tables

**Figure 1 sensors-23-02726-f001:**
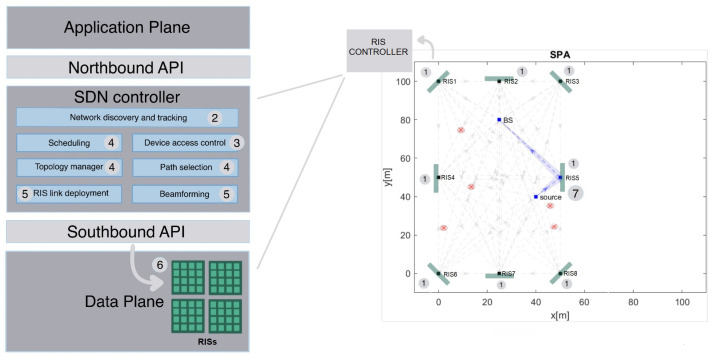
Proposed SDN-oriented system architecture, where the Control Plane is decomposed into functional blocks interacting with the local RIS controller.

**Figure 2 sensors-23-02726-f002:**
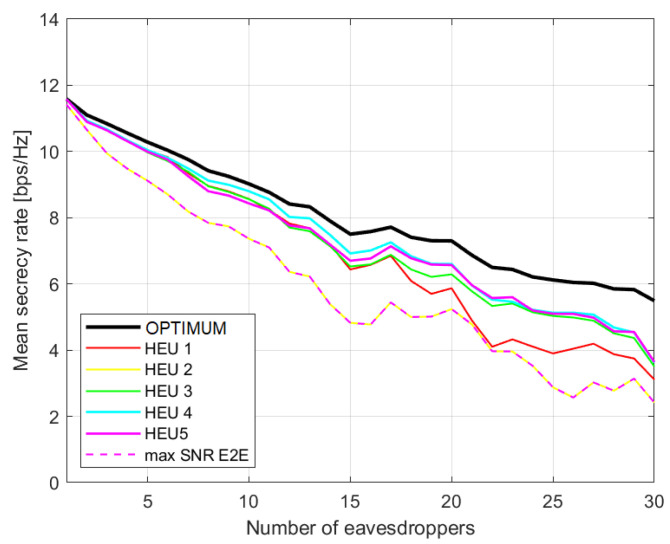
Average secrecy rate with regards to number of eavesdroppers for different approaches. Scenario (c) with cooperative eavesdroppers.

**Figure 3 sensors-23-02726-f003:**
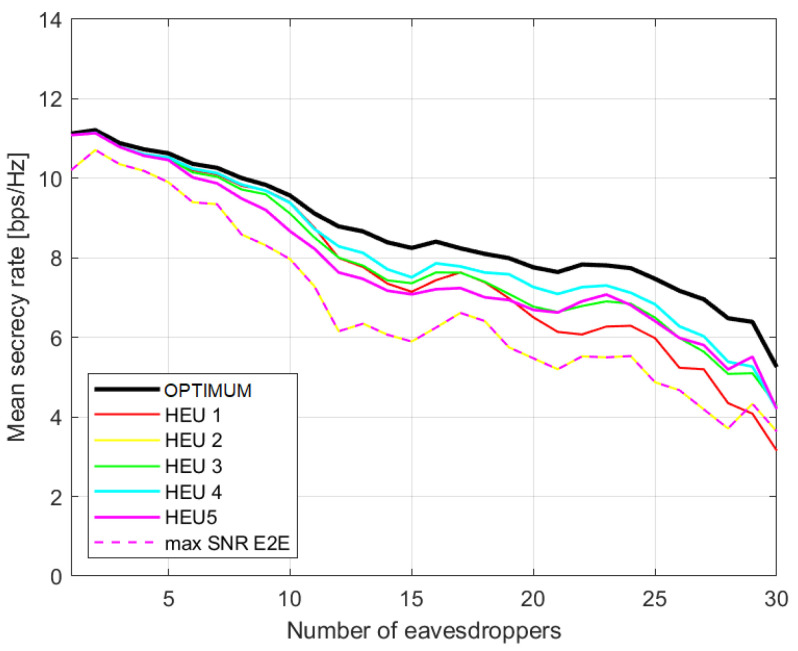
Average secrecy rate with regards to number of eavesdroppers for different approaches. Scenario (nc) with non-cooperative eavesdroppers.

**Figure 4 sensors-23-02726-f004:**
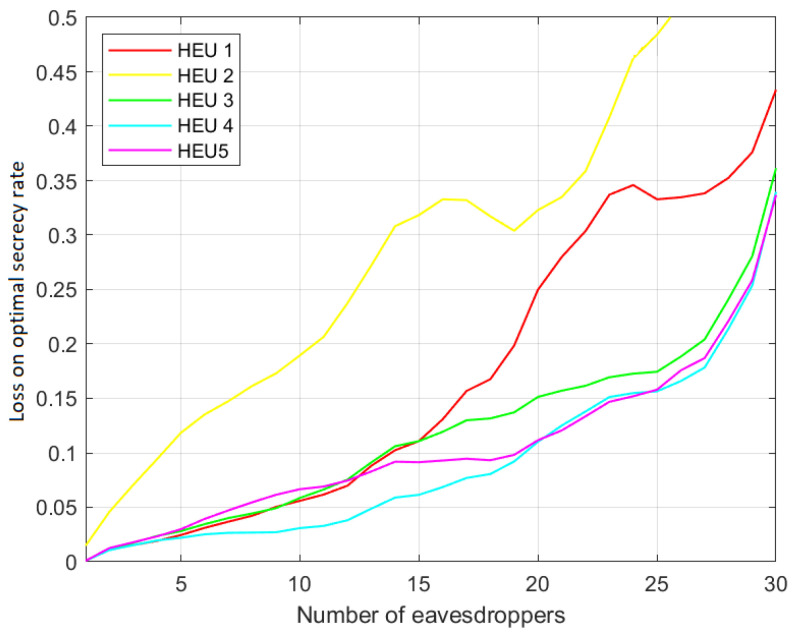
Percentage loss over optimum secrecy rate. Scenario (c) with cooperative eavesdroppers.

**Figure 5 sensors-23-02726-f005:**
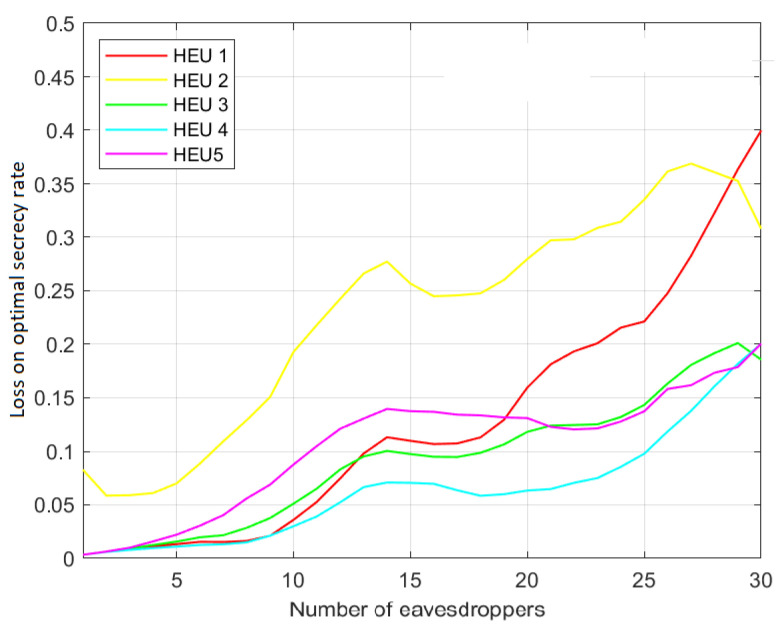
Percentage loss over optimum secrecy rate. Scenario (nc) with non-cooperative eavesdroppers.

**Figure 6 sensors-23-02726-f006:**
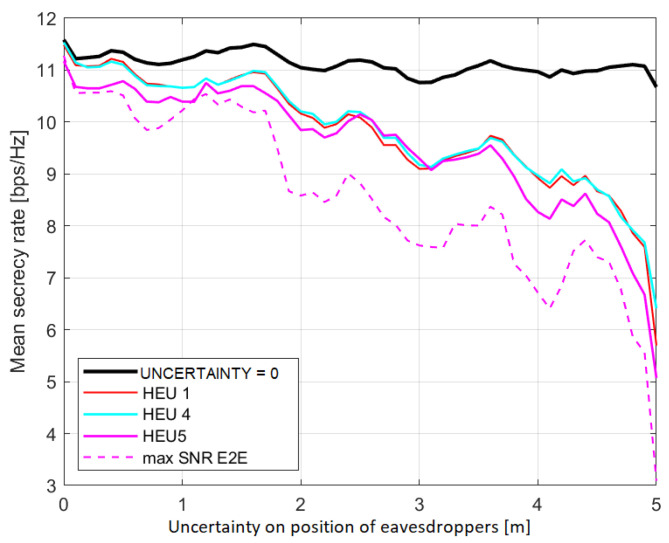
Average secrecy rate versus uncertainty on eavesdroppers’ position.

**Figure 7 sensors-23-02726-f007:**
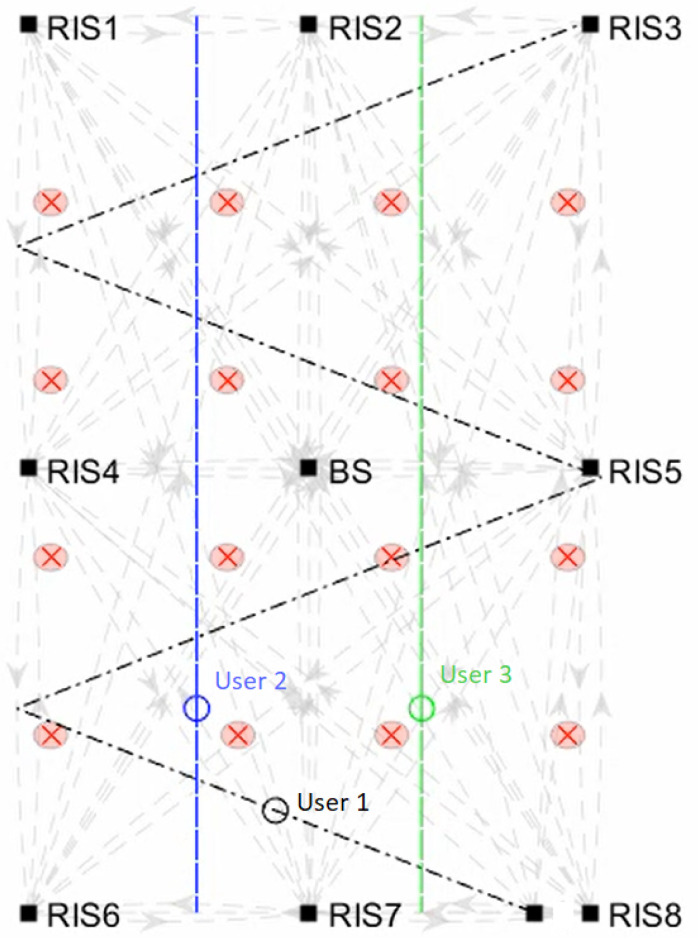
Multi-user composite mobility pattern: two users follow a linear trajectory (blue and green lines), while the third user follows a zig-zag trajectory (black line).

**Figure 8 sensors-23-02726-f008:**
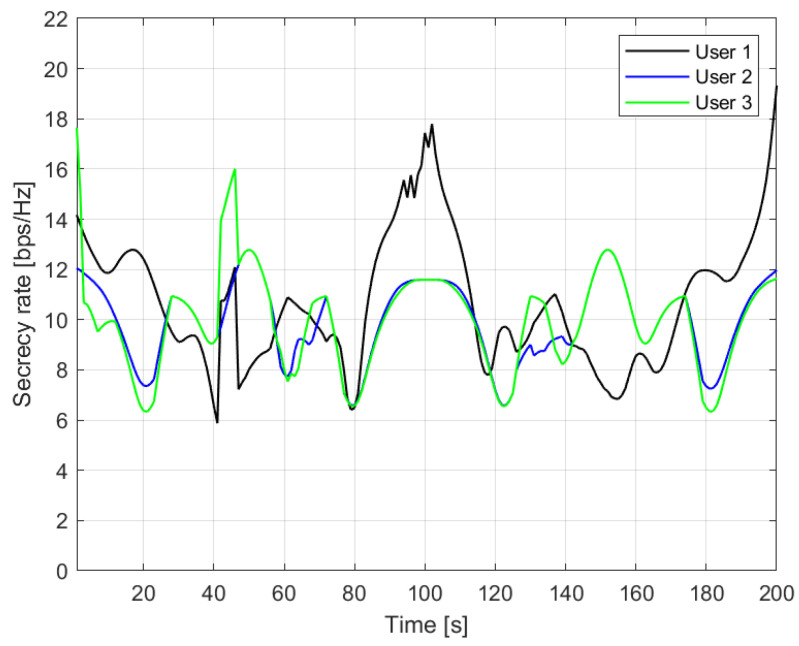
Average secrecy rate versus simulated time.

**Table 1 sensors-23-02726-t001:** Parameters for numerical simulation.

Parameter	Value	Description
*M*	$10^{3}$	Number of reflecting elements
$P_{tx}$	26dBm	Transmit power of user equipment
$\sigma^{2}$	−94dBm	Noise power
$f_{c}$	6GHz	Carrier frequency
ζ	1	In all heuristics except HEU4, where we set ζ=0.55. Reasonable values for our approach, as a result of an empirical optimization.
ρ	0.25	In HEU2, as a reasonable value for our approach, and as a result of an empirical optimization.

## Data Availability

Data sharing not applicable.

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
