# Peer review of "Secure Networking with Software-Defined Reconfigurable Intelligent Surfaces"

_sensors, 2023, doi:10.3390/s23052726_

Round 1

Reviewer 1 Report

The main claim indicated by the autors which is the integration of a multi-RISs system within a SDN architecture is not conveniently presented in the current submitted paper. There are no design of this integration, no implementation details, and no evaluation about this integration. In my opinion, the current paper does not deliver the main purpose indicated by the authors. In addition, the novelty of the current paper in relation to the literature is difficult to identify.

--

It is missing a communication diagram between the SDN controller and the RIS gateway.

What SDN controller algorithm(s) should be run to enable the current proposal?

Fig. 1 uses distinct acronyms (IRS, RSI) for the same technology. I suggest the authors to use a single acronym for that.

I suggest the authors to add in the beginning of Section 4 a Table with all the parameters used in the numerical simulations. The authors should also justify the values assumed for those parameters.

Why the authors used a static (10^3) value for M (number of reflecting elements) in opposition to the variable number used by the literature (e.g. in ref.[9], M varies in range [400,1600])?

Missing solution details:

- how the SDN controller can estimate the eavesdroppers position?

Missing assumptions justifications:

- pg.8, why the users cannot share any Reconfigurable Intelligent Surface (RIS) within their e2e path? What impact this scenario would have on the solution evaluation? 

Missing results:

- show HEU1 and HEU2 are computationally lighter than other proposed alternatives.

- the authors should compare their proposal against the best available proposals (e.g. ref. [9] in the literature).

- the authors should outline the novelties of their proposal in relation to the literature.

The figure 5 about the mobility scenario is hard to interpret. The authors should put it clearer for the reader.

The authors should also justify why they have used this very simplistic mobility pattern to test their proposal.

Some typos to correct:

pg.2,l.61, weighed

pg.6, wieghts

pg.7,l.203, "expect" should be changed to "except"

Reviewer 2 Report

This paper needs the following revision:

1) Please add more simulations

2) Please add a section "Problem statement" to clarify your problem and formulation

3) Please survey more papers on Security for IRS. The following paper should be cited and commented on: Artificial Noise Aided Secure NOMA Communications in STAR-RIS Networks

Reviewer 3 Report

This paper proposed the integration of a multi-RISs system within a Software Defined Networking (SDN) architecture to provide a specific control layer for secure data flow forwarding. The optimization problem is properly characterized in terms of the objective function and an equivalent graph theory model is provided to evaluate the optimal solution. Besides, different heuristics are proposed, trading off the complexity and PLS performance, in order to evaluate the more suited multi-beam routing strategy. The article falls within the scope of the journal.

*I recommend supporting Section 2 "Related works" with new references related to the topic and providing a summary table of all previous works comparing with your current work at the end of this section.

*Simulation setup in Section 4, Paragraph 2 needs to support by reference.

*Section 4 is not enough and needs to support more results and more deep discussion.

*Recommend making proofreading to reduce typos and grammar issues.

Round 2

Reviewer 1 Report

I think the paper quality has improved. Below I have some comments.

The paper should be completely proofreaded. As an example, in the abstract, "in order to evaluate the more suited" should be replaced by "to evaluate the most suitbale"; "which point out" should be changed to ", which points out". The last phrase of the abstract is too long. The authors sould rewrite it in smaller phrases.

Which version of MATLAB  was used (as well as internal used toolboxes)?

I think the caption of Figure 4 is missing the type of tested scenario with (cooperative?) eavesdroppers.

Typos: l324->"trough";

Write in a more clear way: l327 -> "in order to evaluate the more suited multi-beam routing strategy" should be changed to "to evaluate the most suitable multi-beam routing strategy"

The last phrase of "Conclusions" is too long; it should be splitted in smaller phrases.

Reviewer 2 Report

No Comments

Reviewer 3 Report

The authors have addressed the comments
